# Translating Motion to Notation: Hand Labanotation for Intuitive and Comprehensive Hand Movement Documentation

Ling Li
Shenzhen Key Laboratory of
Ubiquitous Data Enabling, Shenzhen
International Graduate School,
Tsinghua University
Shenzhen, China
liling22@mails.tsinghua.edu.cn

WenRui Yang
Shenzhen Key Laboratory of
Ubiquitous Data Enabling, Shenzhen
International Graduate School,
Tsinghua University
Shenzhen, China
ywr22@mails.tsinghua.edu.cn

Junliang Xing
Department of Computer Science and
Technology, Tsinghua University
Beijing, China
jlxing@tsinghua.edu.cn

Xinchun Yu
Shenzhen Key Laboratory of
Ubiquitous Data Enabling, Shenzhen
International Graduate School,
Tsinghua University
Shenzhen, China
yuxinchun@sz.tsinghua.edu.cn

Xiao-Ping Zhang*
Shenzhen Key Laboratory of
Ubiquitous Data Enabling, Shenzhen
International Graduate School,
Tsinghua University
Shenzhen, China
xpzhang@ieee.org

## ABSTRACT

Symbols play a pivotal role in the documentation and dissemination of art. For instance, we use musical scores and dance notation to document musical compositions and choreographic movements. Existing hand representations do not fit well with hand movement documentation since (1) data-oriented representations, e.g., coordinates of hand keypoints, are not intuitive and vulnerable to noise, and (2) the sign language, another widely adopted representation for hand movements, focuses solely on semantic interaction rather than action encoding. To balance intuitiveness and precision, we propose a novel notation system, named Hand Labanotation (HL), for hand movement documentation. We first introduce a new HL dataset comprising 4M annotated images. Thereon, we propose a novel multi-view transformer architecture for automatically translating hand movements to HL. Extensive experiments demonstrate the promising capacity of our method for representing hand movements. This makes our method a general tool for hand movement documentation, driving various downstream applications like using HL to control robotic hands.

## CCS CONCEPTS

• **Applied computing** → **Media arts**.

## KEYWORDS

Hand Labanotation, Movement Documentation

*Corresponding author. Email: xpzhang@ieee.org

*MM '24, October 28-November 1, 2024, Melbourne, VIC, Australia*
© 2024 Copyright held by the owner/author(s). Publication rights licensed to ACM.
ACM ISBN 979-8-4007-0686-8/24/10
https://doi.org/10.1145/3664647.3680568

**ACM Reference Format:**
Ling Li, WenRui Yang, Junliang Xing, Xinchun Yu, and Xiao-Ping Zhang. 2024. Translating Motion to Notation: Hand Labanotation for Intuitive and Comprehensive Hand Movement Documentation. In *Proceedings of Proceedings of the 32nd ACM International Conference on Multimedia (MM '24)Proceedings of the 32nd ACM International Conference on Multimedia (MM'24), October 28-November 1, 2024, Melbourne, Australia.* ACM, New York, NY, USA, 9 pages. https://doi.org/10.1145/3664647.3680568

## 1 INTRODUCTION

In music, people symbolize frequencies into various musical notes, thereby creating the musical staff [4] for the documentation and dissemination of music. In the domain of dance, dance movements are symbolized, resulting in the Human Labanotation for recording dance movements [9, 16, 31]. An intuitive, accurate, and comprehensive method should document the content practically. As shown in Figure 1, symbols replace data-based methods for documenting music and dance movements.

Hand movements are a crucial component of daily human communication and human-computer interaction [23], making their documentation and dissemination equally important. However, existing methods do not adequately fulfill the requirements for documenting hand movements. Data-based recording methods such as Skeleton [8, 28, 34], Mesh [18, 19], and MANO [2, 26], though capable of capturing hand movements with relative precision, are deficient in intuitiveness. Sign Language (SL), as a set of symbols for gestural communication, is likewise unsuitable for recording hand movements. SL as a language focuses more on conveying semantic information than the movements themselves, limiting its ability to cover the full spectrum of hand movements. Additionally, the execution of SL sometimes necessitates the involvement of upper limb movements and facial expressions. Therefore, the sign language notation is not suitable for documenting hand movements.

Inspired by human Labanotation, this paper introduces a novel methodology for documenting hand movements, termed Hand Labanotation. It employs simple symbols to represent the motion

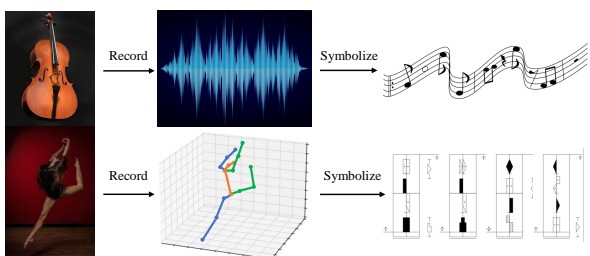

(a) Music and Dance Arts    (b) Data-based Documentation    (c) Symbolized Documentation

**Figure 1: Methods of documenting music and dance movements. (a) Artistic creations such as musical compositions performed by musicians and dance movements executed by humans require documentation and dissemination. (b) Data-based forms allow for accurate recording of results, such as using frequency for music and 3D coordinates of key points for dance movements. (c) Symbolic forms represent the discretization of data-based results, losing a small amount of precision but being more intuitive and easier to disseminate. Existing recordings of music and dance movements often utilize symbolic documentation.**

state of each part of the hand, thereby documenting hand movements. Hand movements revolve around the joints, and based on the hand's physiological structure, we have defined 20 regional vectors based on the hand skeleton model to represent the spatial state of each part of the hand. As illustrated in Figure 2, our framework designs the spatial division method by the activity characteristics of the fingers, resulting in a total of 26 basic Hand Labanotation symbols. To automate the conversion of hand movements into Hand Labanotation scores, we construct a large-scale multiview Hand Labanotation dataset (HLD) with over 4 million annotated hand movement images. We also propose a learning model featuring a novel multiview transformer (MHLFormer) architecture for the automated translation of images into Labanotation scores. To evaluate MHLFormer's performance on HLD, we introduced three evaluation metrics. Furthermore, we propose the Labanotation Hierarchical (LH) Loss Function to facilitate precise model translation. Extensive experimental results validate the accuracy of our method for documenting hand movements. We can also use it as an intermediary representation to control robotic hands.

In summary, our contributions are as follows:

- We introduce Hand Labanotation, a novel symbol system for hand movements, facilitating symbolic documentation and analysis, streamlining hand movement analysis, and applying it to robotic hand control.
- Our creation of the HLD dataset, derived from annotated Interhand2.6M [21] and Freihand [36] data, represents the first extensive multi-view dataset for Hand Labanotation, marking a leap in hand movement documentation.
- We propose and train the MHLFormer model using the innovative LH loss function, achieving outstanding performance in translating hand movements into HL scores.

## 2    RELATED WORK

**Existing Representation Methods for Hand Movement Documentation**. We can broadly categorize feasible methods into three

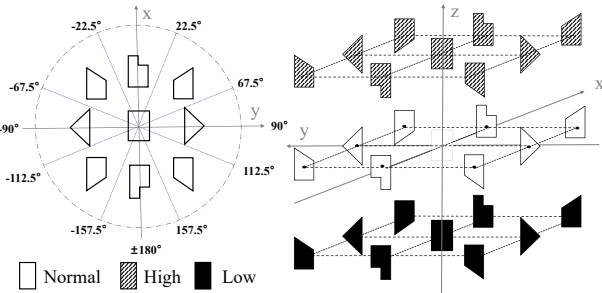

(a) Basic symbols      (b) Spatial mapping of Labanotation symbols

**Figure 2: Fundamental symbols of hand Labanotation and their corresponding spatial representation.**

classes. (1) Using video to record movements. While video recording offers the advantage of being intuitive, it fails to address the issue of occlusions effectively [8, 24]. Additionally, the requirement for substantial storage space is another drawback of video recording [25, 27]. (2) Utilizing data-based hand representation methods for documentation. In computer vision research, 3D hand movement documentation relies on methods such as Skeleton [34, 35], Mesh [18, 19], and MANO [2, 26]. Although these methods can record hand movements with relative accuracy, they severely lack intuitiveness. It is challenging for observers to replicate hand movements merely based on coordinates data. Moreover, capturing accurate hand pose data is also difficult. (3) Employing symbolic methods for documentation. Symbolic recording methods also offer the advantage of being intuitive and, thus, easier to replicate. Existing Sign Language methods [17] have developed their unique system of symbols. However, since Sign Language focuses on semantic information, we cannot document many non-semantic hand movements using Sign Language symbols. Therefore, it is essential to design an intuitive and reliable symbolic system.

**Labanotation for Human Body**. Human Labanotation has a long history. Developed in the early 20th, Labanotation is a symbolic method for recording body movement [9]. With the continuous development of Labanotation, it eventually evolved into a system composed of four basic elements: body, space, time, and dynamics [12]. The body represents the body's moving parts, such as the 'left leg', 'right hand', *etc.*, typically comprising 11 body parts. Space represents the methodology of characterizing movements through direction, level, distance, and degree. Time indicates the duration of each movement, while dynamics denote the emotional components contained within the movement. In the Labanotation score, we describe movements using two dimensions: body and time [15, 31]. The vertical axis represents the temporal progression of the action sequence, with time flowing from the bottom to the top. Each column on the horizontal axis represents a part of the body. The shape and color of symbols represent the spatial position of each body part [9]. With the development of Human Labanotation, the differentiation of human body regions by various Labanotation symbols has become increasingly refined [16, 31]. This refinement provides a viable method for documenting hand movements.

**Motion-to-Labanotation Translation**. The goal of Labanotation translation is to be rapid and accurate. Initially, experts manually recorded Labanotation while observing dance [7, 9]. To streamline this process, computer-assisted tools like Genlaban [6], Laban

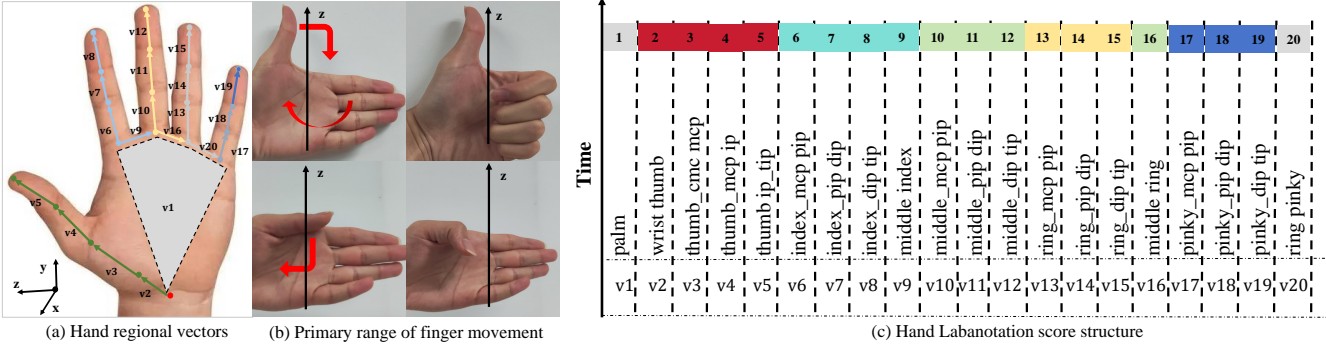

(a) Hand regional vectors    (b) Primary range of finger movement    (c) Hand Labanotation score structure

**Figure 3: The structural composition of Hand Labanotation. (a) The depiction of the coordinate definition and regional vector distribution for the human hand. (b) In our defined coordinate system, the thumb has a larger range of motion along the $Z$-axis direction and a smaller range of motion within the $XY$ plane. Conversely, the remaining four fingers have a larger range of motion within the $XY$ plane and a smaller range along the $Z$-axis. (c) The structure of a single-Hand Labanotation score, with horizontal entries recording 20 regional vectors represented by Labanotation symbols and the vertically upward direction indicating time. We extend the horizontal record to include the other hand's regional vectors for both hands.**

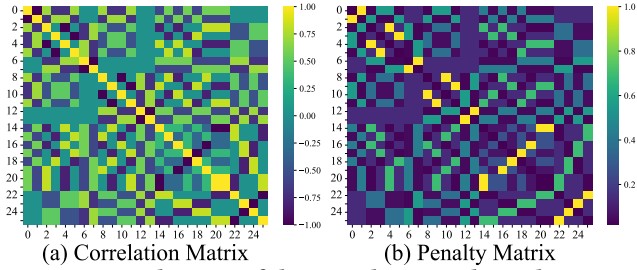

(a) Correlation Matrix    (b) Penalty Matrix

**Figure 4: Visualization of the Correlation and Penalty Matrix for basic HL symbols. The horizontal and vertical coordinates of the image correspond to 26 symbols, respectively. We obtain the correlation matrix for two symbols by calculating the dot product of their direction vectors.**

Writer [3, 30], and Led&Lintel [11, 13] were developed, simplifying symbol creation and speeding up notation. While aiding in Labanotation symbol editing and scoring, these tools demand substantial Labanotation expertise, posing barriers for many users [15, 31]. Recent advances in computer science have led to automated Labanotation translation from 3D motion data, increasing efficiency [31]. This method relies on the accuracy of 3D pose estimation. However, during the pose estimation process, the accuracy of the annotations may be affected due to frequent occlusions. The subsequent manual proofreading workload remains significant. Researchers have developed end-to-end methods and datasets such as Laban16 [15] and Laban48 [31] in response. However, these datasets only cover basic upper limb movements. Lower limb and hand movements tend to be more complex. There is still a lack of existing datasets.

## 3 METHOD

### 3.1 Hand Labanotation Score

Figure 3 displays the correlation of the Hand Labanotation with different parts of the hand. In the following, we will elaborate on the construction process of the Hand Labanotation score.

**Hand Modeling**. The skeletal model of each human hand consists of 21 keypoints based on the hand's skeleton, which we refer to as nodes. As shown in Figure 3 (a), the red node on each hand is located at the wrist, representing the origin of the hand joints.

**Coordinate Setting**. Based on the skeleton model, we can define a coordinate system of the hand. We set the root node as the coordinate origin, let $Y$-axis be in the direction from the root node towards the base of the middle finger, $Z$-axis be in the direction perpendicular to the $Y$-axis and the left on the plane of the palm, and let $X$ be the cross product of $Y$ and $Z$. The hierarchical relationship among the 21 nodes depends on the edge between the nodes and the distance from the joint root and the hand model. For each pair of nodes at the ends of an edge, the node closer to the joint root is regarded as the father of the other.

**Regional Vectors**. From the coordinate information of the nodes, we get 20 vectors $\vec{v}_i, i \in [1, 20]$, which we call regional vectors. Each vector corresponds to a non-root node state, utilizes its father node as the reference, and stands for the node's relative position concerning its father node. These vectors represent the state of the particular section of the hand. Thus, the hand movement is represented by the set $V = (\vec{v}_1, \ldots, \vec{v}_{20})$, as shown in Figure 3 (a). Finally, each regional vector $\vec{v}_i$ can be converted to a symbol in the Hand Labanotation score system based on their spatial positions. In this way, the current hand movement of a single hand can be represented by 20 Hand Labanotation symbols, 40 for both hands. Since the entire palm is close to a rigid structure, we use a region vector parallel to the $y$ axis to represent the whole palm section.

**Basic Hand Labanotation Symbol**. On the surface of a cuboid, we select 8 vertices, the centers of the 6 faces, and the midpoints of the 12 edges. Thus, we obtain 26 points, each corresponding to one of the Hand Labanotation symbols. As shown in Figure 2, we select the center of the cuboid as the coordinate origin, the horizontal plane as the $XY$ plane, and the vertical direction as the $Z$-axis direction. For a given spatial region vector $\vec{v}_i, i \in [1, 20]$, we determine the shape of the symbol based on the projection of $\vec{v}_i$ on the $XY$ plane and the texture of the symbol based on the angle between $\vec{v}_i$ and the $Z$-axis. This way, we can map any spatial region vector to a Hand Labanotation symbol. As shown in Figure 3 (b), the range of motion for the thumb is predominantly along the $Z$-axis.

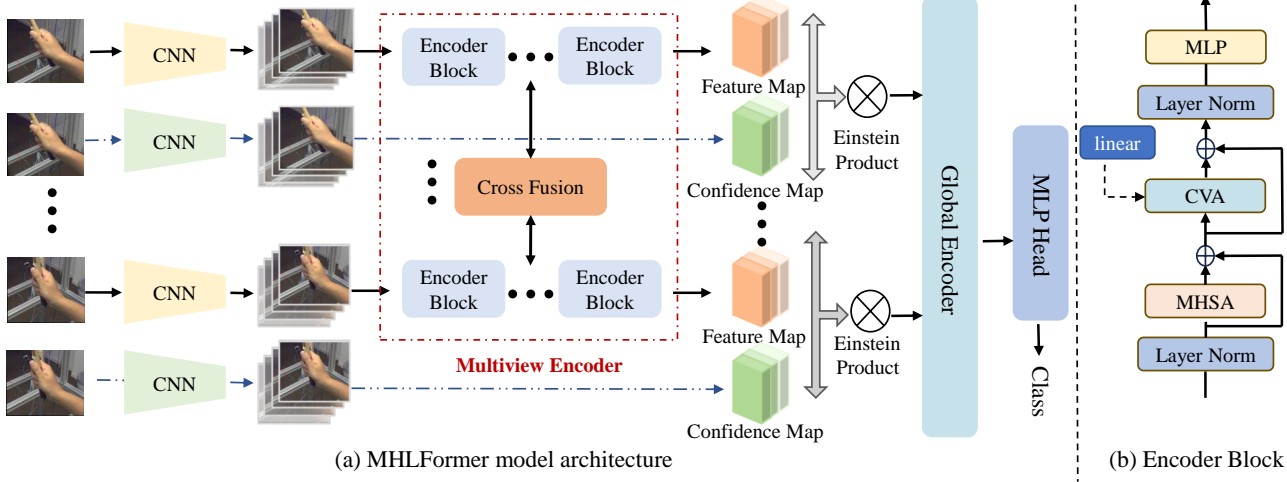

(a) MHLFormer model architecture          (b) Encoder Block

**Figure 5: The architecture of the MHLFormer model. (a) The presentation of the overall structure of the model. We take a series of hand movement images from different views as input. These images first undergo feature extraction through a convolutional neural network (CNN) to obtain feature maps. These maps are then mapped to form a sequence of tokens and further processed through the Encoder Block of the Multiview Transformer for information exchange. Information between adjacent views is fused using a Cross View Fusion model via a linear layer. The Confidence Map (CMP) module, composed of a simple ResNet-50 [10] network, operates parallel to the model to supervise the attention-focused features, with an Einstein product applied to the feature maps at the end. (b) The details of the Encoder Block's structure. The output from the previous view is linked to the Encoder Block of the current view through a linear layer. Here CVA is for Cross View Attention, MHSA is for Multi-Head Self-Attention, and MLP is for multilayer perception. For more details, refer to [22].**

We divide the $Z$-axis direction into 3 intervals. Within the $XY$ plane, the four fingers other than the thumb primarily rotate around the $Z$-axis, so we make a more detailed division of the regions within the $XY$ plane, dividing it into 9 intervals.

## 3.2 Formulation of HL Translation

Due to the reason that the manual drafting of Hand Labanotation scores is both costly and cumbersome, it is necessary to develop a dataset-trained, end-to-end model capable of autonomously translating Hand Labanotation scores for arbitrary hand movements. We formulate this task as follows. The given input data is firstly formulated in the form of $V \in \mathbb{R}^{T \times N \times H \times W \times C}$. $V = \{V_1, V_2, \ldots V_T\}$, where $T$ represents the number of frames of the data. For $\forall i \in [1, T], V_i \in \mathbb{R}^{N \times H \times W \times C}$, which is collected from $N$ cameras located in the same experimental site. $V_i = \{V_i^1, V_i^2, \ldots, V_i^N\}$, for $\forall j \in [1, N], V_i^j \in \mathbb{R}^{H \times W \times C}$ represents the hand movement information at the $i$-th moment from the $j$-th view in the data. Then, we use the function $\mathcal{T}$ to represent the system's transformation from the image data to the translated Hand Labanotation sequence.

$$f : \mathcal{T}(V_i^1, V_i^2, \ldots, V_i^N, \Phi) = L, \qquad (1)$$

where $L = \{p_r^1, p_r^2, \ldots, p_r^{20}; p_l^1, p_l^2, \ldots, p_l^{20}\}$. For $\forall k \in [1, 20], p_r^k$ represents the right part, and $p_l^k$ represents the left part. $p_r^i, p_l^i \in \mathbb{R}^{1 \times cls}$, where $cls = 26$, represents all Hand Labanotation symbol categories that can be used to represent a complete Hand Labanotation space.

## 3.3 Hand Labanotation Dataset

To facilitate the automatic translation of Labanotation scores, we construct a dataset named Hand Labanotation Dataset (HLD) based on InterHand2.6M dataset [21] and Freihand dataset [36]. To our knowledge, it is the first comprehensive dataset of multiview and single-view hand movements. Our dataset encompasses more than 4 million hand movement images, covering a diverse range of hand movements. Specifically, it includes more than 200 in distinct types of hand movements and involves more than 26 in different male and female participants. Each entry is detailed with high-resolution spatial and temporal annotations, capturing nuances in hand movements. The dataset is developed using algorithmic and manual methods to annotate Hand Labanotation. Here, we briefly introduce the HLD annotation process. First, we use rules demonstrated in Sec. 3.1 to calculate all the regional vectors $(\vec{v_1}, \ldots, \vec{v_{20}})$ of each hand. Each of the regional vector $\vec{v} = (x, y, z)$ is then transformed from a Cartesian coordinate system to a spherical coordinate system by the following rule

$$
\begin{aligned}
r &= \sqrt{x^2 + y^2 + z^2}, \\
\theta &= \arccos \frac{z}{r}, \theta \in [0, 180], \\
\Phi &= \arctan(\frac{y}{x}), \Phi \in [-180, 180].
\end{aligned}
\qquad (2)
$$

Identifying the region where the rotation angles $(\theta, \Phi)$ are determined in the corresponding Hand Labanotation symbols. In our dataset, annotation is conducted through a hybrid approach. 19.85% of the joint frames have incomplete information. We manually annotated all joints for these joint frames. For these joints, when the

manually annotated joints with existing information match the algorithmic calculations, we consider the annotation of the missing joint information to be correct. After annotating these frames, we use an algorithm to obtain the results for the remaining frames. The final annotations are preserved as texts. The resultant dataset is categorized into 3 subsets: training, validation, and testing. Figure 6 features a six-frame sequence of a "fist-clenching" gesture captured from multiple views and the corresponding HL symbols.

## 3.4 Multiview Hand Labanotation Transformer

*3.4.1 Cross Fusion Mechanism.* The Cross Fusion mechanism to fuse information from different views is depicted in Figure 5. The overall fusion strategy is designed to draw inspiration from the linear fusion approach in MVT [32]. This entails jointly executing self-attention on a single view. We sequentially blend the information between every two adjacent view pairs to efficiently fuse information while conserving computational resources. Since the number of tokens is the same for each view after encoding, there are no issues related to size matching. The cross-fusion processing between two adjacent views is illustrated as follows

$$\mathbf{z}^{(i+1)} = \text{CVA}\left(\mathbf{z}^{(i+1)}, \mathbf{W}^{\text{proj}}\mathbf{z}^{(i)}\right), \tag{3}$$

where CVA represents the cross-view attention mechanism, which the following formula can express.

$$\text{CVA}(\mathbf{x}, \mathbf{y}) = \text{Softmax}\left(\frac{(\mathbf{W}^Q\mathbf{x})(\mathbf{W}^K\mathbf{y})^\top}{\sqrt{d_k}}\right)(\mathbf{W}^V\mathbf{y}). \tag{4}$$

Consider that $\mathbf{W}^Q, \mathbf{W}^K$, and $\mathbf{W}^V$ represent the query, key, and value projection matrices, respectively, utilized within the attention mechanism [29]. As depicted in Figure 5 (b), $\mathbf{W}^{\text{proj}}$ is the weight matrix after a linear layer during the fusion of adjacent views. Within the Cross-View Attention (CVA) module, the dashed line represents the features of the previous view after passing through the Encoder Block. Furthermore, we incorporate a residual connection surrounding the cross-view attention mechanism.

*3.4.2 Multiview Encoder.* The Encoder Block modules from different views consist of the Multiview Encoder, illustrated in Figure 5 (b). For single frame hand movement, the input data is denoted as $\mathbf{V} \in \mathbb{R}^{N \times H \times W \times C}$, where $N$ denotes the number of views of the input image. We employ ResNet-50 [10] to obtain feature maps. Figure 5 (a) shows that the Encoder Block obtains token sequences by extracting feature maps from images from different views. After passing through the Multiview Encoder block, these token sequences yield feature and confidence maps from different views. They have the same size and hence can be denoted as $\forall i \in [1, N], \mathbf{z_i} \in \mathbb{R}^{d \times cls}$. Here, $d$ denotes the total count of regional vectors associated with a hand movement, while $cls$ signifies the number of fundamental categories within the Hand Labanotation symbol taxonomy. In the Global Encoder module, we concatenate them and then process them through Multi-Head Self-Attention (MHSA) [29]. Finally, we map the encoded classification to the output by a linear classifier.

## 3.5 Labanotation Hierarchical Loss

Since the structure of the hand can be viewed as a tree-like structure emanating from the root node, there is a dependency relationship between different keypoints. We can perform hierarchical supervision based on the structural differences of the keypoints. Accordingly, we introduce the Labanotation Hierarchical (LH) loss to train our model, comprised of HXE loss [1] and penalty loss. We calculate the correlation matrix based on the orientation $\vec{d}$ of different HL symbols in space. Since each symbol corresponds to a portion of the spatial region, we use the direction from the coordinate origin to the center of the spatial region as the direction of the symbol. The calculation process for correlation is as follows

$$Cor[i-1, j-1] = \vec{d}_i \cdot \vec{d}_j, \forall i, j \in [1, 26]. \tag{5}$$

Based on the degree of correlation between different categories in Figure 4, we propose a penalty matrix to generate penalty loss, which assigns weights to various erroneous translations. The penalty matrix is defined as follows

$$M_p = e^{-\alpha(Cor+1)}, \tag{6}$$

where $Cor$ is the correlation matrix between different categories, and $\alpha$ is a positive scaling factor to control the rate at which the penalty coefficient varies with the correlation coefficient. The expression of the penalty loss is

$$\mathcal{L}_{penalty}(\hat{y}, y) = -\sum_{i=1}^{cls}(m_p y_i \cdot \log(m_p \hat{y}_i) + \tag{7}$$
$$(1 - m_p y_i) \cdot \log(1 - m_p \hat{y}_i)).$$

In the above expression, $y$ is the ground truth label, $\hat{y}$ is the raw translation value, and $cls$ represents the number of basic Hand Labanotation symbol categories. $m_p$ is the weight value of the penalty matrix $M_p$ between ground truth and predicted labanotation symbols. Finally, we define the LH loss function of the model as

$$\mathcal{L}_{LH} = \beta\mathcal{L}_{penalty} + (1 - \beta)\mathcal{L}_{\text{HXE}}, \tag{8}$$

where $\beta$ is a hyperparameter ranging between 0 and 1.

# 4 EXPERIMENTS

## 4.1 Dataset

The HLD is constructed based on the Interhand2.6M [21] and Freihand [36] datasets. The HLD dataset consists of $4,243,136$ images, including a training set of $2,525,049$ images, a validation set of $852,343$ images, and a test set of $865,744$ images. A statistical analysis is conducted on Labanotation's distribution across various spaces within all training data subsets for left, right, and both hands. The results are illustrated in Figure 7. We find that the distribution of Hand Labanotation in space is relatively uniform. Figure 6 offers a visualization of the dataset, which displays photographs of hand movement sequences and their corresponding HL scores.

## 4.2 Evaluation Metrics

To assess the efficiency of our network's automatic translation result of Hand Labanotation, it is advantageous to adopt a distance metric to quantify the disparity between the translated Hand Labanotation score and the ground truth Hand Labanotation. To provide a comprehensive evaluation, three metrics are proposed in this work:

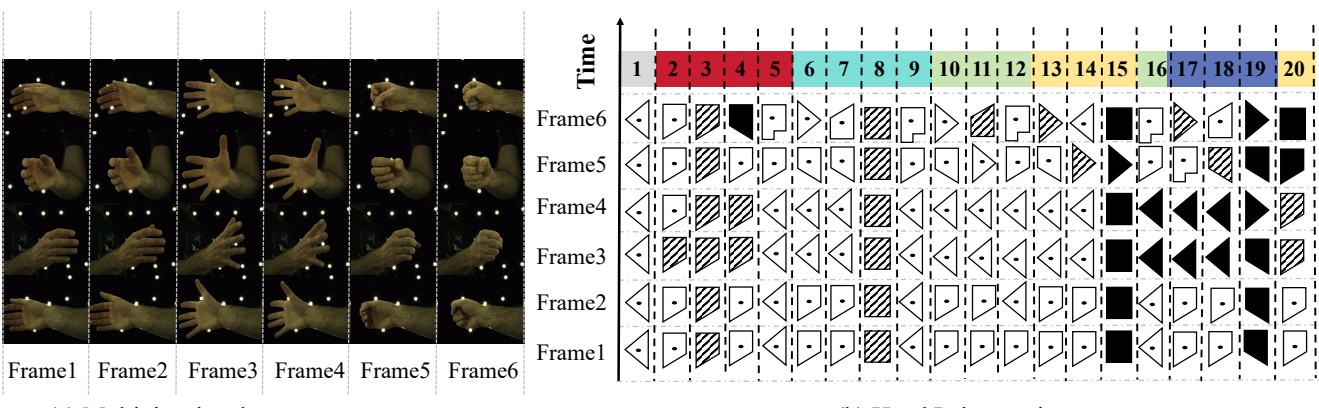

(a) Multiview hand movement sequences          (b) Hand Labanotation scores

**Figure 6: Partial display of dataset data, where one frame of hand movement corresponds to a single row of Hand Labanotation score. Over time, HL scores are generated progressively from bottom to top.**

| Method | $Laban_{acc}$@1 ↑ | $Hor_{acc}$@1 ↑ | $Ver_{acc}$@1 ↑ | Para. (MB) | FLOPs (Giga) |
|---|---|---|---|---|---|
| OpenPose [5] | 46.43% | 54.41% | 75.37% | 36.83 | 50.02 |
| InterNet [33] | 64.72% | 68.89% | 85.44% | 143.37 | 32.81 |
| IntagHand [14] | 59.48% | 65.42% | 76.34% | **39.04** | 17.88 |
| **Ours** | **78.75%** | **81.86%** | **89.35%** | 58.20 | **15.15** |

**Table 1: Comparison of different methods. For fairness, we limit the MHLFormer input to one view.**

$Hor_{acc}$, $Ver_{acc}$, and $Laban_{acc}$. Each metric evaluates performance across the horizontal, vertical, and overall dimensions.

Labanotation symbols translation result can be represented as a matrix $\mathbf{L}$, which is illustrated in Sec. 3.2. The columns correspond to different regional vectors of two hands, and the rows are associated with individual frames. Each element of the matrix indicates the direction of a regional vector. These directions are divided into 26 distinct categories, each corresponding to a specific Hand Labanotation symbol. For $t$ frames data, the translation result from the network of Hand Labanotation is denoted as

$$\mathbf{L} = \begin{bmatrix} p\{t,r_1\} & \cdots & p\{t,r_{20}\} & p\{t,l_1\} & \cdots & p\{t,l_{20}\} \\ \vdots & \vdots & \vdots & \vdots & \vdots & \vdots \\ p\{2,r_1\} & \cdots & p\{2,r_{20}\} & p\{2,l_1\} & \cdots & p\{2,l_{20}\} \\ p\{1,r_1\} & \cdots & p\{1,r_{20}\} & p\{1,l_1\} & \cdots & p\{1,l_{20}\} \end{bmatrix},$$

for $\forall i \in [1, 20], p\{t, r_i\}$ and $p\{t, l_i\}$ represents the transformation result of the Hand Labanotation symbols corresponding to the $i$-th regional vectors of the right and left hands in the hand movement at frame $t$. In our dataset, each sample comprises three distinct components: the left hand, the right hand, and their interaction. Applying a mask matrix, denoted as $\mathbf{M}$, is imperative during the testing phase. The matrix can filter out non-essential elements by using mask weights represented by binary numbers. To evaluate horizontal and vertical Hand Labanotation translation accuracies, we introduced two specific mapping matrices: the horizontal mapping matrix $\mathbf{H}$ and the vertical mapping matrix $\mathbf{V}$. These matrices facilitate the transformation of each Labanotation into forms that exclusively contain horizontal or vertical

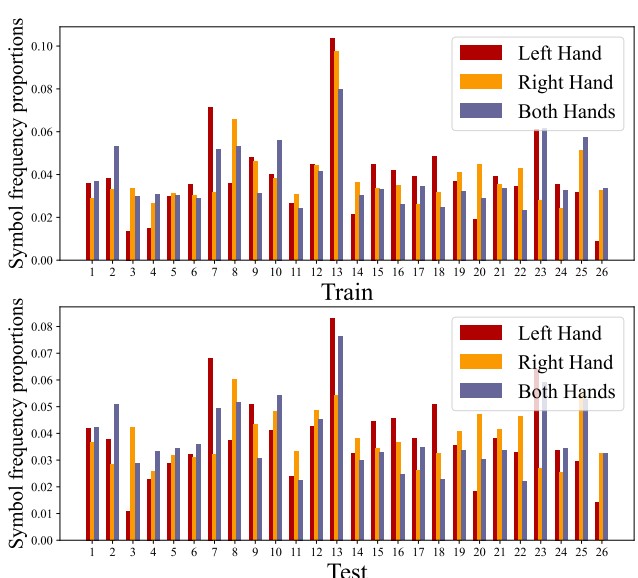

**Figure 7: Statistical distribution of 26 Hand Labanotation symbols in the train and test datasets for the left hand, right hand, and both hands. The visualization demonstrates a relatively balanced quantity of different Labanotation symbols across various data types.**

elements. Building on the ingredients we discussed above, the overall accuracy rate of Labanotation translation is formulated as follows: $Laban_{acc} = \frac{1}{K} \sum_{i=1}^{T} \sum_{j=1}^{M} \delta(\mathbf{L}^{ij}, \mathbf{L'}^{ij}) \cdot \mathbf{M}^{ij}$. Similarly, the accuracy rate of horizontal Labanotation translation is calculated by $Hor_{acc} = \frac{1}{K} \sum_{i=1}^{T} \sum_{j=1}^{M} \delta((\mathbf{L} \cdot \mathbf{H})^{ij}_{pred}, (\mathbf{L'} \cdot \mathbf{H})^{ij}_{gt}) \cdot \mathbf{M}^{ij}$. Lastly, the accuracy rate of vertical Labanotation translation is determined by $Ver_{acc} = \frac{1}{K} \sum_{i=1}^{T} \sum_{j=1}^{M} \delta((\mathbf{L} \cdot \mathbf{V})^{ij}_{pred}, (\mathbf{L'} \cdot \mathbf{V})^{ij}_{gt}) \cdot \mathbf{M}^{ij}$. In these formulas, $K = \sum_{i=1}^{T} \sum_{j=1}^{M} \mathbf{M}^{ij}$, $T$ represents the total number of sample frames, $M = 40$ denotes all hand parts within a single frame, $\mathbf{L}$ and $\mathbf{L'}$ represent the predicted and ground truth Labanotation symbols on each regional vector, respectively, and $\delta(x, y)$ is an

| Backbone | Method | Loss | $Laban_{acc}$@2 ↑ | $Hor_{acc}$@2 ↑ | $Ver_{acc}$@2 ↑ | $Laban_{acc}$@4 ↑ | $Hor_{acc}$@4 ↑ | $Ver_{acc}$@4 ↑ |
|---|---|---|---|---|---|---|---|---|
| ResNet-50 | - | $\mathcal{L}_{CE}$ | 73.85% | 77.34% | 88.59% | 76.66% | 81.01% | 90.10% |
| ResNet-50 | - | $\mathcal{L}_{HXE}$ | 74.23% | 77.36% | 89.95% | 79.89% | 82.13% | 90.33% |
| ResNet-50 | - | $\mathcal{L}_{LH}$ | 75.13% | 79.50% | 91.78% | 82.14% | 86.53% | 92.46% |
| ResNet-50 | Cross Fusion | $\mathcal{L}_{CE}$ | 79.09% | 82.86% | 91.30% | 83.47% | 86.77% | 92.69% |
| ResNet-50 | Cross Fusion | $\mathcal{L}_{HXE}$ | 80.11% | 82.90% | **92.49%** | 85.89% | 87.20% | 94.21% |
| ResNet-50 | Cross Fusion | $\mathcal{L}_{LH}$ | **83.46%** | **85.79%** | 92.09% | **87.71%** | **89.48%** | **95.27%** |
| ResNet-152 | - | $\mathcal{L}_{CE}$ | 76.38% | 80.15% | 89.68% | 77.45% | 82.57% | 90.64% |
| ResNet-152 | - | $\mathcal{L}_{HXE}$ | 76.82% | 79.30% | 90.28% | 77.90% | 82.68% | 90.86% |
| ResNet-152 | - | $\mathcal{L}_{LH}$ | 78.92% | 84.14% | 91.94% | 85.16% | 87.30% | 94.73% |
| ResNet-152 | Cross Fusion | $\mathcal{L}_{CE}$ | 80.98% | 83.71% | 91.51% | 84.88% | 87.72% | 92.79% |
| ResNet-152 | Cross Fusion | $\mathcal{L}_{HXE}$ | 82.64% | 84.22% | 93.76% | 87.45% | 89.63% | 94.94% |
| ResNet-152 | Cross Fusion | $\mathcal{L}_{LH}$ | **84.17%** | **86.36%** | **94.43%** | **88.04%** | **89.99%** | **95.31%** |

**Table 2: A summary of results for MHLFormer using different backbones, with and without the Cross Fusion mechanism, and employing various loss functions. We conducted experiments with** 2 **and** 4 **views, respectively.**

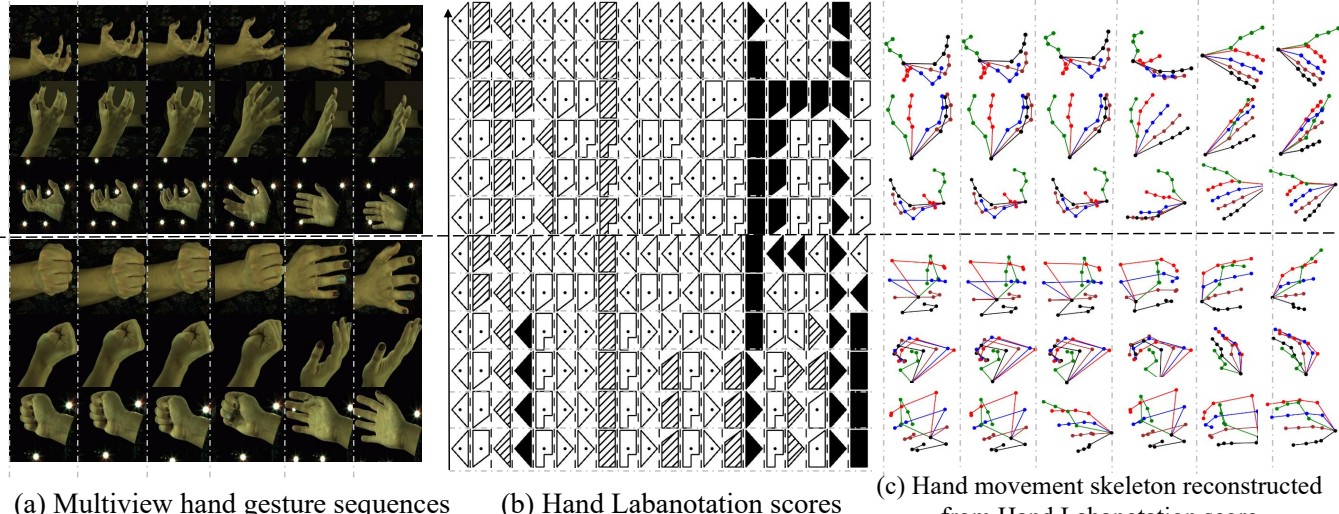

(a) Multiview hand gesture sequences          (b) Hand Labanotation scores          (c) Hand movement skeleton reconstructed from Hand Labanotation score

**Figure 8: Testing from hand movement to Hand Labanotation score and then to the reconstruction of hand movement skeleton. (a) The display of two multiview sequences of hand movements. (b) The translation result of Hand Labanotation scores using MHLFormer. (c) The visualization of the hand movements skeletons reconstructed from the translated HL scores.**

indicator function

$$\delta(x, y) = \begin{cases} 1 & \text{if } x = y \\ 0 & \text{otherwise} \end{cases}. \qquad (9)$$

## 4.3 Comparison with Existing Methods

Hand movement documentation is a novel task, similar to 3D hand pose estimation. We can apply methods from 3D hand pose estimation to our task and compare them with our method. Since our dataset is constructed based on the Interhand2.6M [21] and Freihand [36] datasets, we select models trained on the Interhand dataset and fine-tune them on the Freihand dataset. After the model training is complete, we can obtain HL results by calculating based on the 3D coordinates derived from the 3D pose estimation method. We compare our method with methods OpenPose [5], InterNet [33], and IntagHand [14]. For fairness, we standardize the number of

input image views to one. Thus, we can compare our method with existing 3D hand pose estimation methods. As seen from Table 1, the performance of our model is significantly superior to methods that generate HL based on existing hand pose estimation models. The size and computational demand of our model are also very competitive.

## 4.4 Ablation Study

In our experiments, we compared various loss functions. Table 2 shows that our proposed loss function significantly outperforms the traditional Cross-Entropy (CE) and HXE losses, indicating its superior efficacy with different backbones. Additionally, Table 2 highlights the effectiveness of the cross-fusion module in enhancing model performance, which suggests that it facilitates better information integration across views. Notably, @2 in the table represents the use of 2 views in the experiment.

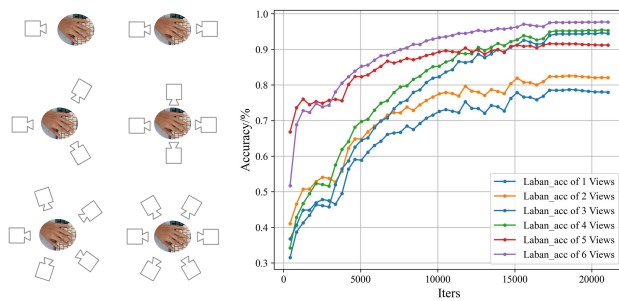

(a) Perspectives selection  (b) Translation accuracy under different perspectives

**Figure 9: Sampling methods of views and the translation accuracy of HL with varied numbers of input viewpoints. Experimental results demonstrate that selecting four views as inputs to the MHLFormer can achieve efficient performance.**

### 4.5 Implementation Details

The Interhand2.6M [21] dataset provides many images captured from multiple views, from which we select various viewpoints. We select 6 multiview schemes in the experiment with 1 to 6 views around the hand. The details can be found in Figure 9 (a). The accuracy of Hand Labanotation translation $Laban_{acc}$ for Hand Labanotation at each iteration is calculated for each scheme. Figure 9 (b) illustrates the $Laban_{acc}$ curves corresponding to the 6 different schemes. It is shown that the scheme with 4 views enables the translation of accurate Hand Labanotation with relatively fewer resources. Correspondingly, in the Freihand [36] dataset, we also select 4 viewpoints from these same positions. We conducted our experiments on 2 Nvidia RTX 3090 GPUs. A total of 50 epochs is trained, and the learning rate is $1 \times 10^{-4}$. Since the model's Backbone does not significantly impact the experimental results, we primarily use the ResNet [10] model in the CNN module.

## 5 DISCUSSION AND APPLICATIONS

Hand Labanotation can be an intermediate representation for downstream tasks, yielding satisfactory results. Herein, we will present some potential applications of HL.

### 5.1 A Control Signal for Robotic Hand

Hand Labanotation, as a symbolic system, effectively applies to documenting hand movements and controlling robotic hands. Our proposed Hand Labanotation symbol quantifies the rotation angles of individual hand vectors in polar coordinates, which shares similarities with the current approach of using dense rotation angle parameters to control robotic hands. Hand Labanotation can directly serve as an intermediary representation for non-high-precision robotic hand movement imitation, efficiently realizing an integrated process from documenting to simulating hand movements. Although HL loses some precision regarding angles, it also acts as a natural filter for angles, which can prevent severe jitter. Experimental results indicate that HL can replicate human hand movements with relative accuracy as a language for controlling robotic hand actions. Our method is faster and more efficient than capturing images with optical cameras and then using a 3D hand pose estimation model to estimate hand coordinates for controlling a robotic hand.

### 5.2 An Efficient Representation for ASL

Sahand [20] is a dataset comprising $64,000$ images of American Sign Language (ASL), covering 10 numbers and 26 English alphabet. The essence of this task is a gesture classification problem with 36 categories. Initially, after pre-training the IntagHand model [14] on the Interhand2.6M dataset [21], we predict the 3D gesture representation from input images, including Skeleton, MANO, and Mesh. Subsequently, we directly connect the results to the final output through the Linear layer. Similarly, we employ the MHLFormer model, pre-trained on the HL dataset, to predict HL representations. Based on HL, we also use the Linear layer for gesture classification.

### 5.3 A Symbol System for Approximate Gesture Recovery

To further explore the rationality of Hand Labanotation, we converted the predicted Labanotation back to the hand skeleton. After initializing the root node coordinates, the rotational information of the hand is restored based on the Hand Labanotation symbols for each segment. We represent the spatial orientation with the central axis of the space corresponding to HL symbols. By assigning phalange lengths proportional to those of an adult hand, we obtain the skeleton results as shown in Figure 8. A comparison with the original input images reveals that while not completely precise, the skeleton structure reconstructed from Hand Labanotation still accurately records the hand movements relatively well. For example, the two actions shown in Figure 8 are both hand-opening movements, and the results after reconstruction still allow for an intuitive understanding of the corresponding actions.

Based on the aforementioned capabilities and the distinct differences in shape and texture of HL symbols, HL can be used as Braille to encode gestures, further expanding its application scenarios.

## 6 CONCLUSION

This paper introduces Hand Labanotation, a symbolic system designed to document hand movement sequences intuitively. The expansive Hand Labanotation dataset (HLD), a comprehensive repository comprising over 4 million frames capturing single and multiview hand movements, accompanies this novel notation system. Furthermore, we propose the MHLFormer method, a pioneering network specifically designed for the autonomous translation of hand movements into HL symbols. Experimental results demonstrate that our method achieves commendable performance. We achieve relatively good results using the predicted HL for hand movement reproduction. HL can also directly serve as an intermediary representation for controlling robotic hands, thereby unifying documentation and dissemination.

**Limitations:** Due to the loss of some information in the process of symbolization, there will be a small amount of unavoidable errors when we store hand movements with HL and then restore them.

**Acknowledgements**: This research of Hand Labanotation is supported in part by the Shenzhen Ubiquitous Data Enabling Key Lab under grant ZDSYS20220527171406015, and by the Tsinghua Shenzhen International Graduate School-Shenzhen Pengrui Endowed Professorship Scheme of the Shenzhen Pengrui Foundation.

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
