# OpenReview forum: "Translating Motion to Notation: Hand Labanotation for Intuitive and Comprehensive Hand Movement Documentation"
_acmmm.org/ACMMM/2024/Conference — MM2024 Oral_

### Official Review · Reviewer_U6Bc · 2024-05-16

**Rating:** 5
**Confidence:** 3

**Summary:**

This paper introduces Hand Labanotation (HL), an innovative notation system for hand movement documentation. To help convert hand movements into HL, the authors have constructed a large-scale multiview Hand Labanotation dataset (HLD) with over 4 million annotated hand movement images. With this dataset, they further propose a novel multiview transformer (MHLFormer) architecture for the automated translation of images into Labanotation scores. Some novel metrics and loss functions are applied to facilitate the training and evaluation of the framework. This is a meaningful exploration and also helps promote the development of various downstream tasks.

**Strengths:**

1. The proposed Hand Labanation is interesting, which utilizes simple symbols to represent the movements of the human hand.
2. A dataset consisting of Interhand2.6M and Freihand datasets was constructed, with over 4 million images labeled with corresponding symbols.
3. Furthermore, the authors also propose a novel multiview transformer (MHLFormer) architecture to translate the input images into Labanotation scores. Meanwhile, both evaluation metrics and Labanotation Hierarchical (LH) Loss Function are proposed to facilitate the translation.

**Limitations:**

1. I suggest the author adjust Figure 1 so that it can visually display the proposed tasks.
2. For section4.4, please supplement more analysis and explanation, for example, analyzing why the proposed loss is more effective.
3. I suggest the author place the section4.5 at the beginning of Chapter 4, which may be more convenient for readers to follow this paper.
4. Both the InterNet and IntagHand in Table 1 are mainly designed for modeling interacting hands. For those single hand frames, I think comparing them with the SOTA methods of single hand modeling is more convincing.
5. Although the constructed dataset reach up to 4 million, as mentioned in main paper, it comes from existing data and the authors re-labeling them to obtain  HLD. So using it as a key contribution may be relatively fragile.

**Suitability:**

3

---

### Official Review · Reviewer_Hrqn · 2024-05-23

**Rating:** 4
**Confidence:** 3

**Summary:**

This work introduces Hand Labanotation (HL), a novel system for documenting hand movements. The authors contributed the HLD dataset and proposed a multi-view transformer architecture to automatically translate hand movements into HL. The paper also provides potential applications and evaluations on downstream tasks.

**Strengths:**

This work has several strengths:

(1) It introduces a novel notation system for documenting hand movements, which is an intriguing task.

(2) It proposes the MHLFormer model, which uses the LH loss function to achieve great performance in translating hand movements into HL scores.

(3) Additionally, this work contributes a multi-view dataset for Hand Labanotation.

**Limitations:**

(1) In the related work, it says, “Sign Language cannot capture the spatial details of a wide range of hand movements…” near line 169. Please provide further support for this claim.

In American Sign Language (ASL), there are five parameters to convey meaning: handshape, palm orientation, movement, location, and expression/non-manual signals. The sign language book provides detailed text descriptions on how to correctly make sign language in spatial space. For ASL analysis methods like 3D landmarks or mesh rotation matrices, numbers are used to indicate the spatial details. In contrast, employing discrete symbolic Hand Labanotation (HL), which clusters some spatial spaces into one symbol, affects the accuracy of representing spatial details.

(2) One of the novel aspects mentioned in the MHLFormer model architecture is the Multiview Encoder. However, at line 694, it mentions standardizing the number of input image views to one. Please provide more details about this process: do you select one specific view branch or simply duplicate it multiple times?

(3) Alphabet recognition is the simplest task in ASL, yet in Table 3, this work still does not achieve better accuracy than current methods. Although the storage is efficient, accuracy is the first priority in conveying meaning. More experiments on word-level or sentence-level ASL are needed to support your claim near line 800.

This method might be helpful in encoding handshapes but is questionable in ensuring the correct semantic recognition or translation in ASL. For example, in ASL, the same handshape against the chin means "mother," while against the forehead it means "father." This work likely cannot cover such nuances well.

**Suitability:**

3

---

### Official Review · Reviewer_3tkq · 2024-05-25

**Rating:** 3
**Confidence:** 2

**Summary:**

The paper proposes the use of Labanotation, a symbolic representation of body movements, for hand annotation. To do so, it describes the proposed hand notation system and transformation between the orientation of the edges among hand joints (called regional vectors) and the symbols, and proposes a novel large-scale, multiview dataset of hand movement videos annotated with the proposed Labanotation symbols, which is based on existing hand datasets Interhand2.6M and Freihand. The paper also proposes a methodological baseline based on the existing multi-view transformer (MVT) to convert from images to symbols directly. It also proposes to use a combination of the existing hierarchical cross-entropy loss along with a cross-entropy loss informed with the correlation of different symbols.

Evaluation includes: a comparison with some 3D pose estimation methods (e.g. OpenPose), from which the labanotation symbols can be extracted based on the proposed transformation process, showcasing the improved performance of the proposed method; a comparison among different hand pose representation methods (e.g. MANO) with respect to accuracy and storage, showing competitive performance. The paper also performs a series of ablations, including different CNN backbones and the impact of the number of views used on performance. Finally, the proposed representation is preliminary validated in three applications: a control signal for a robotic hand, a representation for ASL, and symbol-to-gesture recovery, demonstrating that the symbolic representation is resource-efficient despite losing some information detail in the process.

**Strengths:**

- The paper deals with an interesting problem and, to my knowledge, this is the first time that Labanotation is proposed for the representation of hand poses.
- The problem is properly motivated and contextualized.
- The hand labanotation score is explained clearly and detailedly.
- The created dataset greatly contributes to fostering this new research topic.
- The images throughout the paper are very informative.
- The ablations are mostly exhaustive and provide rich insights beyond reporting accuracy.
- Despite not achieving the highest accuracy among other hand representation methods, it achieves the goal of efficient storage.
- Having preliminary results for three downstream applications is also greatly appreciated, making the paper complete.

**Limitations:**

The main limitation I’ve encountered is the lack of clarity in the methodological part (sections 3.3. to 3.5), and part of the evaluation (Sec 4.2) hindering its understanding:
- L387-L389 and L433-L437: it is unclear to me how the dataset is annotated from these explanations. Is it manually, semi-automatically, completely automated? And how are the annotations achieved for each case?
- Sec. 3.4. starts directly with the cross-fusion mechanism, without an introduction to the overall pipeline. It feels as if such introduction is just included in the caption of Figure 5 to save space, but that impacts the paper negatively.
- In Sec 3.4.2, there are some parts that are different from Figure 5. For instance, it is said the Encoder block extracts features, when that is supposedly done with ResNet/CNN blocks. Also, it is said that the confidence maps are given after passing through the Multiview encoder block, but in the figure they appear to come from some CNNs.
- Similarly, the pipeline is unclear. In Figure 5, each view is apparently processed by two different CNNs (with different colors), but that is not explained in the text. The output of one goes to the encoder block while the other’s output apparently directly creates a confidence map. How is this confidence map created? Also, according to the text, the CNNs are used to extract features, but Figure 5 depicts the features as ‘multiple’ images. Is this a visualization issue or is there something else?
- Sec 3.4.2 mentions that the last layer is a linear classifier (L488), but the representation is d (20 regional vectors) x cls (26 possible symbols), and the loss in Eq. 7 is a binary cross-entropy-like loss. It is unclear to me how these three items are compatible with each other. What is the format of the final network predictions, and how is the loss computed?
- In Eq. 7, it is also unclear to me how the penalty term is applied and its effect.
- L618 mentions M for the first time, being ‘imperative’ because ‘The matrix can filter out non-essential elements by using mask weights represented by binary numbers.’ However, the text does not describe which type of elements have to be filtered and why.

In addition, the text claims that the paper proposes a novel multi-view transformer, but the only difference I can see compared to the original MVT is the use of a confidence map, which is not justified either.

With respect to the comparison to other methods, I wonder why it was not possible to compare to existing methods for Labanotation, like [15] and [31]. Even if they are originally considered for body notation, they could be extended to hand notation to have a more fairer comparison.

Other comments:
- Using the same math symbols for different elements throughout the text can be confusing, e.g. ‘i’ takes different meanings in L274, and Eq.3. (Also, in Eq. 3 ‘i’ is not defined).
- The current text lists figures unorderedly. For instance, Figure 4 is located on page 3, but not mentioned until page 5.
- For completeness, the HXE loss should be defined.
- In Figure 9 (b), It is surprising that using 5 views the performance saturates, being worse than using 3 or 4. And using 6 works better than 5. This suggests that the issue is not the number of views specifically, but their location.

For these reasons, I believe the paper is not ready to be published in its current form, hence my rating of ‘borderline reject’. But as mentioned, most concerns are due to a lack of understanding so those may be addressed in the rebuttal.

**Suitability:**

2

---

### Official Review · Reviewer_ihNc · 2024-05-26

**Rating:** 5
**Confidence:** 4

**Summary:**

In this paper, a hand labanotation (HL), a new notation system for hand  movement documentation is proposed. The authors construct  a new HL dataset with 4M annotated images. A multi-view transformer architecture is proposed. The comparison results to the state-of-the-art methods make this paper convincing.

**Strengths:**

The hand labanotation score issue proposed in 3.1 and Fig. 3 is well defined. The system architecture in Fig. 5 is clear and readable. The hand labanotation dataset proposed in Sec. 3.3 and the cross fusion mechanism proposed in Sec. 3.4 are reasonable. The comparison results shown in Table 1 demonstrate that the proposed method achieved the best result in most aspects. Teh FLOPs of the proposed method achieved the best result. The results in Table 2 and Table 3 make this paper more convincing. The multiview results in Fig. 9 are convincing for the possible readers.

**Limitations:**

The proposed method not achieving the smallest para. in Table1. The results in Fig. 7 has higher values in the case 13, more discussions are experiments are needed.

**Suitability:**

3

---

### Meta-Review · Area_Chair_ydB7 · 2024-07-06

**Recommendation:** Accept (Oral)
**Confidence:** 5

**Metareview:**

This paper received four borderline/weak acceptances from the reviewers. Overall, all reviewers acknowledged the potential of the proposed symbolic representation, the new dataset for hand movement research, and the MHLFormer model. The reviewers raised concerns regarding the clarity of the descriptions and figures, which should be straightforward for the authors to address.